# Assessing the impact of institutional mistrust on parental endorsement for COVID-19 vaccination among school communities in San Diego County, California

**Tina Le , Marlene Flores, Vinton Omaleki, Ashkan Hassani, Anh V. Vo, F. Carrissa Wijaya, Richard S. Garfein , Rebecca Fielding-Miller \***

University of California San Diego, Herbert Wertheim School of Public Health and Human Longevity Science, La Jolla, California, United States of America

\* rfieldingmiller@health.ucsd.edu

## Abstract

### Background

Institutional mistrust has weakened COVID-19 mitigation efforts. Assessing to what extent institutional mistrust impacts parental decision making is important in formulating structural efforts for improving future pandemic response. We hypothesized that institutional mistrust is associated with lower parental endorsement for COVID-19 vaccination.

### Methods

We distributed an online survey among parents from schools in areas with high levels of social vulnerability relative to the rest of San Diego County. We defined vaccination endorsement as having a child aged 5 years or older who received at least one COVID-19 vaccine dose or being very likely to vaccinate their child aged 6 months—4 years when eligible. Institutional mistrust reflected the level of confidence in institutions using an aggregate score from 11 to 44. We built a multivariable logistic regression model with potential confounding variables.

### Findings

Out of 290 parents in our sample, most were female (87.6%), reported their child as Hispanic/Latino (73.4%), and expressed vaccination endorsement (52.1%). For every one-point increase in mistrust score, there was an 8% reduction in the likelihood of participants endorsing vaccination for their child. Other statistically significant correlates that were positively associated with vaccination endorsement included parent vaccination status, child age, parent age, and Hispanic/Latino ethnicity.

### Conclusion

Our study further demonstrates how institutional mistrust hinders public response during health emergencies. Our findings also highlight the importance of building confidence in

**Data Availability Statement:** Study data will be publicly available through the University of

California San Diego Library Research Data Curation Repository. https://library.ucsd.edu/research-and-collections/research-data/index.html.

**Funding:** Funding for this study was provided by the County of San Diego Health and Human Services Agency (https://www.sandiegocounty.gov/hhsa/) and the National Institutes of Health (https://www.nih.gov/). Funding was awarded to RFM (NIH grant numbers K01MH112436, U01HD108787). The funders had no role in study design, data collection and analysis, decision to publish, or preparation of the manuscript.

**Competing interests:** The authors have declared that no competing interests exist.

institutions and its downstream effects on pandemic preparedness and public health. One way that institutions can improve their relationship with constituents is through building genuine partnerships with trusted community figures.

## Background

Widespread misinformation, social upheaval, and political polarization during the pandemic weakened government credibility and institutional trust [1, 2]. Overall confidence in major US institutions (e.g., government, law enforcement, organized religion, schools) declined between 2020–2022 to its lowest point on record [3, 4]. Perceived competence of local elected leaders and public health officials has fallen to 50% favorability in the United States, down from 79% since the initial outbreak in early 2020 [5]. Consequently, only 25% of adults believe that the US is very prepared to deal with future COVID-19 variant outbreaks [6]. Trust is essential for successfully navigating through public health emergencies [7], but the current trust deficit hinders public cooperation.

Institutional mistrust is a lack of confidence toward a particular organization, reflecting doubt or skepticism about the institution's trustworthiness [8]. Institutional mistrust is dynamic, determined by how well an entity performs based on the expectations of its constituents [9]. When an institution is no longer perceived as trustworthy, its constituents are less receptive to its actions, which threatens pandemic response strategies. For example, trust in government sources of information for vaccines declined during the pandemic and varied by sociopolitical climate and geographic location [6, 10]. This helped exacerbate health disparities by putting marginalized communities, who already have limited access to healthcare resources, at further risk of infection [11, 12].

Institutional mistrust has been strongly associated with lower vaccine uptake among adults during pandemics [13–15]; however, little data exists on the association between institutional mistrust and vaccination coverage among children [16]. Children are at high risk of disease transmission in schools and are especially vulnerable to the social, economic, and developmental consequences of a pandemic [17]. Assessing to what extent institutional mistrust impacts parental decision making is important in formulating structural efforts for improving pandemic response. We sought to test the hypothesis that institutional mistrust is associated with lower parental endorsement for child COVID-19 vaccination. We defined parental vaccination endorsement as having a child aged 5–17 years who received at least one dose of a COVID-19 vaccine or being very likely to vaccinate their child aged 0–4 years once the COVID-19 vaccination became available for their age group.

## Methods

We conducted a secondary analysis of data from a cross-sectional survey nested within the Safer at School Early Alert (SASEA) Program, an environmental monitoring system aimed at preventing COVID-19 outbreaks in childcare and K-8 school sites across San Diego County, California [18].

### Setting and participants

San Diego is the second most populous county in California and is ethnically diverse, with about 23% of residents identifying as an immigrant and 35% identifying as Hispanic or Latino [19]. Approximately 5% f the population is undocumented, which is greater than the national

average [20, 21]. SASEA school sites were selected for participation if they had elevated COVID-19 case rates and were located in census tracts with high levels of social vulnerability according to the California Healthy Places Index [22]. At the time of the study, children aged 5–17 years were authorized for vaccination from the Food and Drug Administration.

All parents and guardians of SASEA-affiliated students were eligible for this study. To avoid overburdening parents with surveys, three classrooms were randomly selected per school site for each monthly survey wave. Parents were recruited through paper flyers in English and Spanish that were sent home with students. Classroom teachers also sent out email announcements with virtual flyers to encourage participation.

## Data collection

Monthly surveys were sent to parents of children enrolled at SASEA sites asking about their perceptions and experiences during the pandemic regarding topics such as healthcare use, perceived wellbeing, and engagement in COVID-19 mitigation behaviors. Surveys were self-administered using REDCap, a web-based survey tool provided by the University of California, San Diego [23, 24]. Participants could access the survey by scanning a quick response (QR) code provided on the recruitment flyer with their smartphone or by calling a study phone number to complete the survey with the assistance of an English-Spanish bilingual researcher. Survey items covered demographic data and perceptions about physical, mental, and social health during the pandemic; no identifiable information was collected. Surveys were offered in English and Spanish and were distributed in two waves between February 7 and April 11, 2022. If a classroom had been sampled twice, responses from the survey wave with the lower response rate were excluded from analysis.

## Variables and measures

Our outcome of interest for this analysis was parental vaccination endorsement. We measured this variable by combining responses for two survey items: "Has your child [aged 5–17 years] received at least one dose of a COVID-19 vaccine?" and "How likely is your child [aged 4 years and under] to get an approved COVID-19 vaccine when it becomes available?" Skip logic was used in the survey to present the appropriate question after the respondent specified their child's age. Responses were categorized as *"Yes"* if the participant indicated that their child received at least one dose or that their child was very likely to get the vaccine when it became available. Responses were categorized as *"No"* if the participant indicated that their child had not received at least one dose or that their child was fairly likely, not too likely, or definitely not likely to get the vaccine when it became available.

Our primary predictor was institutional mistrust, measured by the survey item: "Please indicate how much confidence you, yourself, have in the following institutions: your church, government officials, public schools, newspapers, pharmaceutical companies, television news, the police, news websites, the U.S. Immigration and Customs Enforcement (ICE) and Customs and Border Patrol (CBP), the County Board of Supervisors, and UC San Diego." and quantified trust using a four-point likert-like scale ranging from *A great deal (4)* to *Not at all (1)*. Total mistrust index scores could range from 11 to 44, with 11 representing the lowest mistrust score and 44 representing the highest mistrust score. This survey item was adapted from the Gallup Poll's Measurement for Confidence in Institutions [25].

Potential confounders included in the analysis were parent vaccination status, household income, age of the child who brought the survey flyer home, parent education level, parent age, parent gender, and child ethnicity. Parent vaccination status was a binary variable that reflected whether the respondent had received any dose of a COVID-19 vaccine. Child's

ethnicity was a binary variable that asked whether or not the participant's child was of Hispanic or Latino origin. We used this variable to reflect the large Hispanic/Latino population in our study population. Household income was determined by self-reported household gross income earned in 2019. Parent education level was determined by the respondent's self-reported highest level of education completed. Both income and education levels were measured as an ordinal variable but were treated as continuous to acknowledge that these variables are a spectrum in real life and to avoid misclassification bias.

## Statistical analysis

We conducted bivariate analyses using chi-square for categorical variables and t-test for continuous variables to assess the association between child vaccination status, institutional mistrust, and potential confounders. We then built logistic regression models to test our hypothesized association between institutional mistrust and child vaccination status after adjusting for confounders.

Missing data was treated as missing completely at random after conducting sensitivity analyses and was handled using listwise deletion. Responses were also stratified by children aged 0–4 years and children aged 5 years or older to assess differences in correlates in the logistic regression model. Statistical significance was defined as p<0.05. RStudio version 4.2.0 was used for analysis.

## Ethics

This study was approved by the UC San Diego Human Research Protections Program with protocol number 201627. School principals and district leaders were continuously involved in the dissemination of this program. As part of the survey, participants were asked to review and sign a consent form before proceeding with the survey items. Participation in this survey was voluntary and participants could skip questions they did not want to answer. As an incentive to increase response rates, all respondents were entered into a raffle to win a nominal prize after completing the survey. Study findings were summarized and shared with participants at the conclusion of the school year.

## Results

In total, 507 individuals completed the survey between February and April 2022, 290 respondents of these provided responses for all relevant items and were included in this analysis (Fig 1). Most participants were female (87.6%), had at least a high school education (86.6%), and reported that their child was Hispanic/Latino (73.4%) (Table 1). Approximately half of participants reported an annual household income of $35,000 or less in 2019. Eighty-eight percent of participants reported receiving at least one dose of a COVID-19 vaccine while 52.1% of participants endorsed COVID-19 vaccination for their child. The mean institutional mistrust index score was 26.4 with a standard deviation of 6.3. Sensitivity analysis revealed that there were no significant differences in sociodemographic characteristics between the included and excluded participants (analysis not shown).

Parents who endorsed vaccination reported significantly higher confidence in institutions than parents who did not endorse vaccination. The mean institutional mistrust index score was 3.2 points lower for parents who endorsed vaccinating their child compared to parents who did not endorse vaccination (p<0.0001) (Table 2). Parents who endorsed vaccination had significantly higher rates of COVID-19 vaccination coverage than parents who did not endorse vaccination (97.7% vs. 77.7%, p<0.0001). Average reported child age and parent age were significantly greater among participants who endorsed vaccination compared to those who did

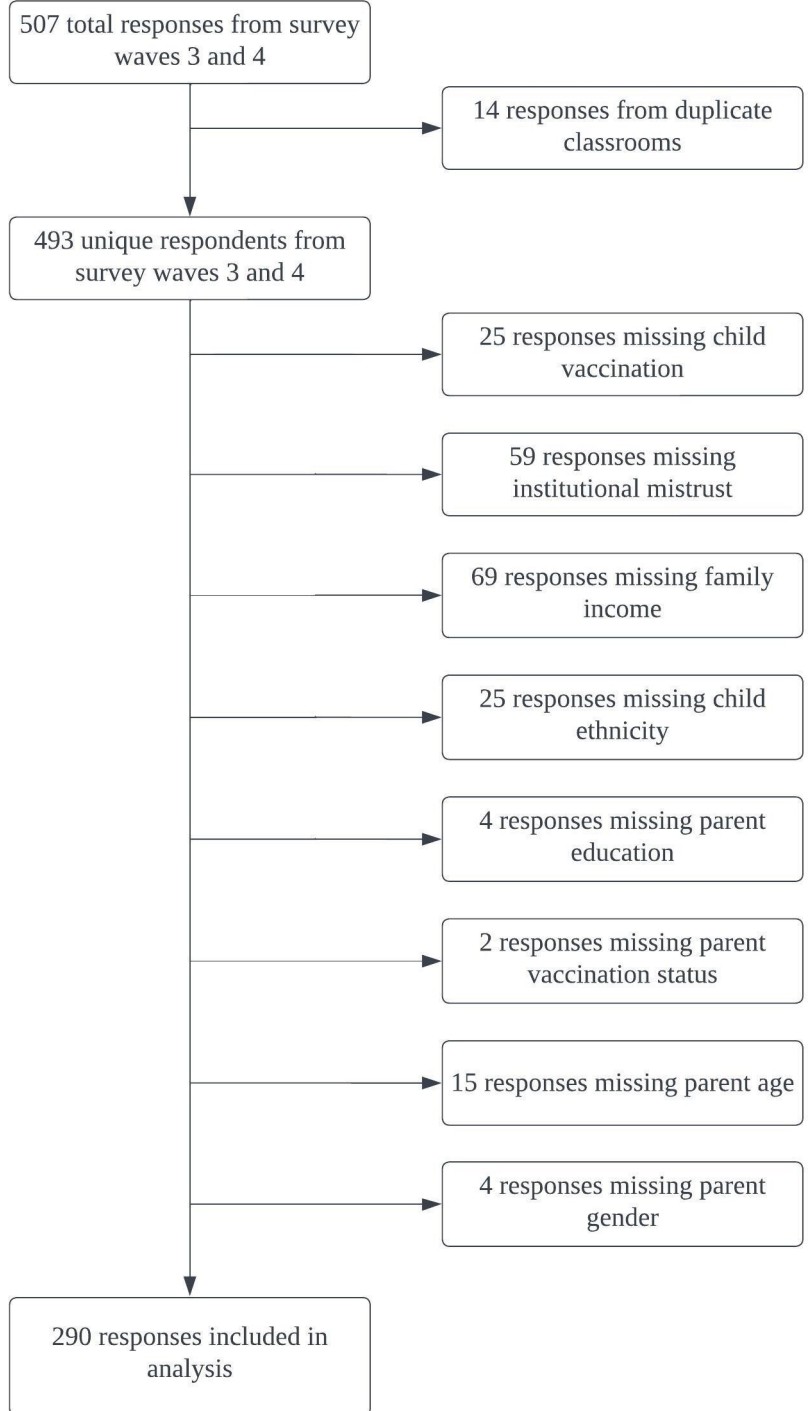

**Fig 1. Flow diagram for selection of survey respondents.**

not (8.5 years vs. 7.7 years [p = 0.0049]; 38.9 years vs. 35.3 years [p = 0.00017], respectively). There was a significant difference in mean family income between parents who endorsed vaccination and unendorsed vaccination (p = 0.0039), but no significant difference in mean parent education level, proportion of Hispanic/Latino ethnicity, and proportion of female identifying respondents.

**Table 1. Survey respondent characteristics (n = 290).**

| Variable | n (%) |
|---|---|
| *Institutional Mistrust Index (Mean±SD)* | 26.4±6.2 |
| *Parent Age (Mean±SD)* | 37.2±8.4 |
| *Child Age (Mean±SD)* | 8.1±2.6 |
| *Parental Vaccination Endorsement* | |
| No | 139 (47.9) |
| Yes | 151 (52.1) |
| *Parent Gender* | |
| Male | 36 (12.4) |
| Female | 254 (87.6) |
| *Child Ethnicity* | |
| Not Hispanic/Latino | 77 (26.6) |
| Hispanic/Latino | 213 (73.4) |
| *Parent Vaccination Status* | |
| Not Vaccinated | 33 (11.4) |
| Vaccinated | 257 (88.6) |
| *Annual Family Income* | |
| <15,000 | 42 (14.5) |
| 15–20,000 | 22 (7.6) |
| 20–25,000 | 33 (11.4) |
| 25–35,000 | 44 (15.2) |
| 35–50,000 | 49 (16.9) |
| 50–75,000 | 50 (17.2) |
| 75–100,000 | 27 (9.3) |
| >100,000 | 23 (7.9) |
| *Parent Education Level* | |
| Less than High School | 39 (13.4) |
| High School or Equivalent | 157 (54.1) |
| Bachelor's Degree | 72 (24.8) |
| Postgraduate Degree | 22 (7.6) |

After adjusting for covariates, higher mistrust scores were associated with lower odds of COVID-19 vaccine endorsement (Table 3). For every one-point increase in mistrust score, there was an 8% reduction in the likelihood of participants endorsing vaccination for their child (95% CI: 0.88–0.96). This association remained statistically significant after stratifying by ages 0–4 and ages five and older. Parent vaccinated status had the largest effect size on vaccination endorsement (aOR 16.5, 95% CI: 4.6–105.8). Hispanic/Latino ethnicity (OR 2.6, 95% CI: 1.3–5.0), income (aOR 1.23, 95% CI: 1.06–1.44), child age (aOR 1.16, 95% CI: 1.04–1.29), and parent age (aOR 1.05, 95% CI: 1.01–1.09) were also positively associated with vaccine endorsement. Parent education level and female gender were not significantly associated with vaccination endorsement after adjusting for covariates.

## Discussion

Our study found a statistically significant association between institutional mistrust index scores and parental vaccination endorsement after adjusting for potential confounders. Higher mistrust index scores were associated with lower odds of COVID-19 parental vaccination endorsement among parents affiliated with SASEA. These findings highlight an important

**Table 2. Bivariate analysis of factors associated with parental endorsement for COVID-19 Vaccination.**

| | Vaccination Endorsement | | |
| | No | Yes | |
| **Variable** | **n (%)** | **n (%)** | **p-value** |
| --- | --- | --- | --- |
| *Mistrust Index (Mean±SD)* | 28.1±6.0 | 24.9±6.0 | <0.0001 |
| *Parent Vaccination Status* | | | <0.0001 |
| No | 31 (22.3) | 2 (1.3) | |
| Yes | 108 (77.7) | 149 (97.7) | |
| *Ethnicity* | | | 0.11 |
| Not Hispanic or Latino | 43 (30.9) | 34 (22.5) | |
| Hispanic or Latino | 96 (69.1) | 117 (77.5) | |
| *Family Income* | | | 0.0039 |
| <15,000 | 22 (15.8) | 20 (13.2) | |
| 15–20,000 | 15 (10.8) | 7 (4.6) | |
| 20–25,000 | 19 (13.7) | 14 (9.3) | |
| 25–35,000 | 20 (14.4) | 24 (15.9) | |
| 35–50,000 | 26 (18.7) | 23 (15.2) | |
| 50–75,000 | 23 (16.5) | 27 (17.9) | |
| 75–100,000 | 8 (5.6) | 19 (12.6) | |
| >100,000 | 6 (4.3) | 17 (11.3) | |
| *Child Age (Mean±SD)* | 7.7±2.4 | 8.5±2.8 | 0.0049 |
| *Parent Age (Mean±SD)* | 35.3±6.7 | 38.9±9.3 | 0.00017 |
| *Education Level* | | | 0.18 |
| Some High School | 15 (10.8) | 24 (15.9) | |
| High School or Equivalent | 89 (64.0) | 68 (45.0) | |
| Bachelor's Degree | 27 (19.4) | 45 (29.8) | |
| Postgraduate Degree | 8 (5.6) | 14 (9.3) | |
| *Parent Gender* | | | 0.66 |
| Male | 16 (11.5) | 20 (13.2) | |
| Female | 123 (88.5) | 131 (86.8) | |

**Table 3. Univariate and multivariable logistic regression analysis of factors associated with parental vaccination endorsement for COVID-19.**

| Variable | Crude Odds Ratio (95% CI) | Adjusted Odds Ratio (95% CI) |
| --- | --- | --- |
| Institutional Mistrust Index | 0.91 (0.87–0.95) | 0.92 (0.88–0.96) |
| Parent Vaccination Status | | |
| No | 1 (ref) | 1 (ref) |
| Yes | 21.38 (6.29–133.85) | 16.49 (4.61–105.78) |
| Ethnicity | | |
| Not Hispanic or Latino | 1 (ref) | 1 (ref) |
| Hispanic or Latino | 1.54 (0.91–2.62) | 2.55 (1.33–4.99) |
| Family Income | 1.18 (1.05–1.32) | 1.23 (1.06–1.44) |
| Child Age | 1.14 (1.04–1.25) | 1.16 (1.04–1.29) |
| Parent Age | 1.06 (1.03–1.10) | 1.05 (1.01–1.09) |
| Education Level | 1.22 (0.91–1.65) | 1.05 (0.70–1.57) |
| Gender | | |
| Male | 1 (ref) | 1 (ref) |
| Female | 1.17 (0.58–2.40) | 0.74 (0.33–1.66) |

opportunity for improving vaccine distribution efforts toward children. Current polls estimate that approximately 50% of parents of children aged 5–17 years and 80% of parents of children under 5 years of age report delaying or refusing to utilize the COVID-19 vaccine for their child [26]. Despite all age groups having been approved for vaccination, institutional mistrust remains a roadblock for vaccine uptake. Successful vaccination programs are contingent on people's trust in the institutions and actors that develop and administer the programs [27].

In our study, Hispanic/Latino children were twice as likely to be vaccinated compared to non-Hispanic/Latino children. This is a stark difference to national vaccination rates, where 48% of Hispanic/Latino children aged 5–17 were vaccinated in April 2022 compared to 73% of Hispanic/Latino children in our study [28]. These results reflect the strong collective efforts to reach Spanish speaking populations in San Diego County during the vaccine rollout. By July 2020, 60% of COVID-19 cases with known race/ethnicity in San Diego were Hispanic/Latino despite only making up 34% of the total population [29]. By 2021, there had been a major shift in leadership and perspective in San Diego County. The County Board of Supervisors had declared racism a public health crisis and had sworn in their first binational Latina leader [30, 31]. County officials partnered with local organizations and worked to expand testing in areas with large Spanish speaking populations, train community health workers (promotores) to educate communities, and launch ads and public awareness campaigns on popular Spanish media sites [32, 33]. By May 2021, 50% of Hispanic/Latino residents in San Diego County had received their first dose of a COVID-19 vaccine as compared to 25% in the US [34].This exemplifies how successful partnerships with the community of interest can overcome social and structural barriers to health.

When institutional mistrust is high, organizations can improve their trustworthiness by fostering collaboration with trusted stakeholders and aligning themselves with the interests and goals of their constituents [35, 36]. These practices promote genuine investment and inclusion of both the institution and its constituents in policy making, providing services, and other processes. Likewise, clear and transparent communication from trusted figures within the community should focus on not only promoting health messaging but also building confidence in the institutions. These actions are crucial for rebuilding trust in institutions and improving public health.

Our data should be interpreted in the context of some study limitations. Most of the participants in our sample reported that they were vaccinated, which may have overestimated the association between parental vaccination status and child vaccination endorsement; however, we propose that this limitation is minimal because vaccination coverage in our study population was similar to the county. We asked participants to report their household income earned in 2019 which may not reflect their financial situations during the pandemic. Similarly, the survey item captured income in a variety of intervals, and we were unable to determine family income relative to participant household size, which may have overestimated the association between family income and vaccination endorsement. We also did not ask participants their reason for not vaccinating their child, which may have omitted potential confounding variables that were not included in analysis.

## Conclusion

In the context of the COVID-19 pandemic, parents must navigate through unprecedented issues and determine the best courses of action for their children's health. The rise of misinformation, political polarization, and social upheaval has complicated pandemic response and has contributed to institutional mistrust. Our study found that institutional mistrust is associated with lower odds of parental endorsement for vaccination against COVID-19. A way to

mitigate institutional mistrust is through building genuine community partnerships and disseminating culturally sensitive care and resources. Findings from this paper highlight the importance of building confidence in institutions and its downstream effects on pandemic preparedness and public health.

## Acknowledgments

We would like to thank our school communities for allowing us to work with them. Thank you to all the principals, teachers, students, and parents who made SASEA possible.

## Author Contributions

**Conceptualization:** Rebecca Fielding-Miller.

**Data curation:** Marlene Flores, Vinton Omaleki, Ashkan Hassani, Anh V. Vo.

**Formal analysis:** Tina Le.

**Funding acquisition:** Rebecca Fielding-Miller.

**Investigation:** Marlene Flores, Vinton Omaleki, Ashkan Hassani, Anh V. Vo.

**Methodology:** Richard S. Garfein, Rebecca Fielding-Miller.

**Project administration:** F. Carrissa Wijaya.

**Supervision:** F. Carrissa Wijaya, Richard S. Garfein, Rebecca Fielding-Miller.

**Visualization:** Richard S. Garfein, Rebecca Fielding-Miller.

**Writing – original draft:** Tina Le.

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
