## [Decision Letter · Decision Letter 0]

14 Sep 2023

PONE-D-23-15837Assessing the impact of institutional mistrust on parental endorsement for COVID-19 vaccination among school communities in San Diego County, CaliforniaPLOS ONE

Dear Dr. Le,

Thank you for submitting your manuscript to PLOS ONE. After careful consideration, we feel that it has merit but does not fully meet PLOS ONE’s publication criteria as it currently stands. Therefore, we invite you to submit a revised version of the manuscript that addresses the points raised during the review process. The authors should provide the reference or baseline level for the variables in Table 3, particularly for the categorical variables. Please submit your revised manuscript by Oct 29 2023 11:59PM. If you will need more time than this to complete your revisions, please reply to this message or contact the journal office at plosone@plos.org. Please include the following items when submitting your revised manuscript:A rebuttal letter that responds to each point raised by the academic editor and reviewer(s). You should upload this letter as a separate file labeled 'Response to Reviewers'.A marked-up copy of your manuscript that highlights changes made to the original version. You should upload this as a separate file labeled 'Revised Manuscript with Track Changes'.An unmarked version of your revised paper without tracked changes. You should upload this as a separate file labeled 'Manuscript'.If applicable, we recommend that you deposit your laboratory protocols in protocols.io to enhance the reproducibility of your results. Protocols.io assigns your protocol its own identifier (DOI) so that it can be cited independently in the future. For instructions see: https://journals.plos.org/plosone/s/submission-guidelines#loc-laboratory-protocols. Additionally, PLOS ONE offers an option for publishing peer-reviewed Lab Protocol articles, which describe protocols hosted on protocols.io. Read more information on sharing protocols at https://plos.org/protocols?utm_medium=editorial-email&utm_source=authorletters&utm_campaign=protocols.

We look forward to receiving your revised manuscript.

Kind regards,

Sampson Twumasi-Ankrah, PHD

Academic Editor

PLOS ONE

Journal Requirements:

Reviewers' comments:

Reviewer's Responses to Questions

**Comments to the Author**

1. Is the manuscript technically sound, and do the data support the conclusions?

Reviewer #1: Yes

Reviewer #2: Partly

2. Has the statistical analysis been performed appropriately and rigorously? 

Reviewer #1: Yes

Reviewer #2: Yes

3. Have the authors made all data underlying the findings in their manuscript fully available?

Reviewer #1: Yes

Reviewer #2: Yes

4. Is the manuscript presented in an intelligible fashion and written in standard English?

Reviewer #1: Yes

Reviewer #2: Yes

5. Review Comments to the Author

Reviewer #1: The use of secondary data in this study gives an interesting view. Mainly, primary studies have been used in such studies.

The background of this study was however scanty and could not give a better explanation to the issues of misinformation.

Reviewer #2: Review of Manuscript: PONE-D-23-15837

In my opinion, there are a few things to consider to improve the manuscript

Major:

1. The authors determining the reasons or factors that may have prevented them from vaccinating their children or influence them in future vaccinations would have made a lot of difference. Even though this was stated in the study’s limitation, providing such information would have played a key role in making the final conclusion of the study very specific, confidently attributing the outcome of the study with institutional mistrust.

Minor:

2. The authors indicate obtaining a signed consent form from study participants. This should be clear, whether it was also part of the online survey or a separate written consent.

6. PLOS authors have the option to publish the peer review history of their article (what does this mean?). If published, this will include your full peer review and any attached files.

Reviewer #1: No

Reviewer #2: No

---

## [Author Response · Author response to Decision Letter 0]

18 Oct 2023

1. The use of secondary data in this study gives an interesting view. Mainly, primary studies have been used in such studies. The background of this study was however scanty and could not give a better explanation to the issues of misinformation.

a. Thank you for your comment. The authors agree that misinformation is a big influence on COVID-19 vaccination uptake and institutional mistrust; however, sources of misinformation was not a primary outcome of the parent study. The study team made the decision to omit this type of item to reduce burden on study participants. Future research can build off from our study and assess this relationship. 

2. The authors determining the reasons or factors that may have prevented them from vaccinating their children or influence them in future vaccinations would have made a lot of difference. Even though this was stated in the study’s limitation, providing such information would have played a key role in making the final conclusion of the study very specific, confidently attributing the outcome of the study with institutional mistrust.

a. The authors agree that this information would have been very insightful for our analysis; however, as the reviewer points out in the above comment, this was a secondary analysis of survey data collected for a different study objective. Despite this, we believe that our findings are still relevant and important for public health action. Future research can build off from our study and explore individual motivations for COVID-19 vaccination for children.

3. The authors indicate obtaining a signed consent form from study participants. This should be clear, whether it was also part of the online survey or a separate written consent.

a. Thank you for the feedback. We have edited a sentence in the Ethics section to clarify this. The sentence reads, “As part of the survey, participants were asked to review and sign a consent form before proceeding with the survey items.”

4. The authors should provide the reference or baseline level for the variables in Table 3, particularly for the categorical variables.

a. Thank you for the feedback. We have edited the table so that it shows the reference group for categorical variables.

---

## [Decision Letter · Decision Letter 1]

27 Nov 2023

Assessing the impact of institutional mistrust on parental endorsement for COVID-19 vaccination among school communities in San Diego County, California

PONE-D-23-15837R1

Dear Dr. Le,

We’re pleased to inform you that your manuscript has been judged scientifically suitable for publication and will be formally accepted for publication once it meets all outstanding technical requirements.

Kind regards,

Sampson Twumasi-Ankrah, PHD

Academic Editor

PLOS ONE

Additional Editor Comments (optional):

Reviewers' comments:

Reviewer's Responses to Questions

**Comments to the Author**

1. If the authors have adequately addressed your comments raised in a previous round of review and you feel that this manuscript is now acceptable for publication, you may indicate that here to bypass the “Comments to the Author” section, enter your conflict of interest statement in the “Confidential to Editor” section, and submit your "Accept" recommendation.

Reviewer #1: All comments have been addressed

Reviewer #2: All comments have been addressed

2. Is the manuscript technically sound, and do the data support the conclusions?

Reviewer #1: Yes

Reviewer #2: Yes

3. Has the statistical analysis been performed appropriately and rigorously? 

Reviewer #1: Yes

Reviewer #2: Yes

4. Have the authors made all data underlying the findings in their manuscript fully available?

Reviewer #1: Yes

Reviewer #2: Yes

5. Is the manuscript presented in an intelligible fashion and written in standard English?

Reviewer #1: Yes

Reviewer #2: Yes

6. Review Comments to the Author

Reviewer #1: Authors have responded to my earlier comments in the first review. The inputs have been made in the main manuscript. This makes the manuscript more comprehensive and can be well appreciated by the scientific community.

Reviewer #2: The authors have addressed all comments, including how informed consent were obtained from study participants.

7. PLOS authors have the option to publish the peer review history of their article (what does this mean?). If published, this will include your full peer review and any attached files.

Reviewer #1: No

Reviewer #2: No

---

## [Editor Report · Acceptance letter]

15 May 2024

PONE-D-23-15837R1 

PLOS ONE

Dear Dr. Fielding-Miller, 

I'm pleased to inform you that your manuscript has been deemed suitable for publication in PLOS ONE. Congratulations! Your manuscript is now being handed over to our production team.

Kind regards, 

on behalf of

Dr. Sampson Twumasi-Ankrah 

Academic Editor

PLOS ONE